# Epidemiology of the Acceptance of Anti COVID-19 Vaccine in Urban and Rural Settings in Cameroon

**DOI:** 10.3390/vaccines11030625

**Published:** 2023-03-10

**Authors:** Cecile Ingrid Djuikoue, Rodrigue Kamga Wouambo, Majeste Mbiada Pahane, Blaise Demanou Fenkeng, Cedric Seugnou Nana, Joelle Djamfa Nzenya, Flore Fotso Kamgne, Cedric Ngalani Toutcho, Benjamin D. Thumamo Pokam, Teke Apalata

**Affiliations:** 1Department of Public Health, Faculty of Health Sciences, Université des Montagnes, Bangangte BP 208, Cameroon; 2Foundation Prevention and Control, Bangangte BP 208, Cameroon; 3American Association of Microbiology (ASM), ASM Cameroon, Bangangte P.O. Box 222, Cameroon; 4Division of Hepatology, Department of Medicine II, Leipzig University Medical Center, University of Leipzig, 04109 Leipzig, Germany; 5Institute of Fisheries and Aquatic Sciences, University of Douala, Douala P.O. Box 24157, Cameroon; 6Faculty of Medicine and Pharmaceutical Sciences, University of Douala, Douala Box 24157, Cameroon; 7Faculty of Health Sciences, University of Buea, Buea P.O. Box 24157, Cameroon; 8Faculty of Health Sciences, Walter Sisulu University, Mthatha 5117, South Africa

**Keywords:** vaccination, acceptance, COVID-19, epidemiology, Cameroon, urban area, rural area

## Abstract

The COVID-19 pandemic rapidly evolved in December 2019 and to prevent its spread, effective vaccines were produced and made available to the population. Despite their availability so far in Cameroon, the vaccination coverage remains low. This study aimed at describing the epidemiology of the acceptance of vaccines against COVID-19 in some urban and rural areas of Cameroon. A cross-sectional, descriptive and analytical survey was conducted from March 2021 to August 2021 targeting unvaccinated individuals from urban and rural area. After receiving appropriate administrative authorizations and an ethical clearance from the Institutional Review Board (or Ethics Committee) of Douala University (N° 3070CEI-Udo/05/2022/M), a cluster sampling at many degrees was performed and a language-adapted questionnaire was completed by each consenting participant. Data were analyzed using Epi info version 7.2.2.6 software and for *p*-values < 0.05, the difference was considered as statistically significant. Out of 1053 individuals, 58.02% (611/1053) participants were residing in urban and 41.98% (442/1053) in rural areas. Good knowledge relative to COVID-19 was significantly higher in urban areas as compared to rural areas (97.55% vs. 85.07, *p* < 0.000). The proportion of respondents who intended to accept the anti COVID-19 vaccine was significantly higher in urban areas than rural areas (42.55% vs. 33.26, *p* = 0.0047). Conversely, the proportion of anti COVID-19 reluctant respondents thinking that the vaccine can induce a disease was significantly higher in rural areas than urban areas (54 (35.07 vs. 8.84, *p* < 0.0001). The significant determinants of anti-COVID-19 acceptance were the level of education (*p* = 0.0001) and profession in the rural areas (*p* ≤ 0.0001), and only the profession (*p* = 0.0046) in the urban areas. This study globally showed that anti-COVID-19 vaccination remains a major challenge in urban as well as rural areas in Cameroon. We should continue sensitizing and educating the population about vaccine importance in preventing the COVID-19 spread.

## 1. Introduction

COVID-19 is an emerging viral infectious disease caused by the novel coronavirus 2 of the severe acute respiratory system, SARS-CoV 2. The WHO learned of its existence on 31 December 2020, when an outbreak of “viral pneumonia” cases was announced in the city of Wuhan, Hubei province in China [1]. On 30 January, the WHO declared the outbreak a public health emergency of international concern and then a pandemic the 11 March 2020 [2]. On 11 February, the WHO officially named the disease “coronavirus disease (COVID-19)” after germ isolation and molecular characterization on 7 January 2020 [2,3].

As of 27 November 2022, the world recorded 637 million confirmed cases and 6.6 million deaths globally [4]. In general, the United States of America (USA) represents so far the most affected country in the world with above 98,972,375 cases [4]. In Africa, COVID-19 affected all 47 African region countries with 8.887.814 cumulative cases, which represented around two percent of the infections around the world [5]. South Africa was the most drastically affected country, with more than 3.6 million infections, followed by Cameroon with 123,993 cases of COVID-19, of which 1965 died and over 2960 were cured [5].

In view of the rapid transmission capacity of this disease, unspecified interventions such as social distancing, barrier measures and quarantine can slow the spread of the virus and flatten the epidemic curve; however, the COVID-19 epidemic will only end if herd immunity is well established among the population, which is usually acquired through infection or vaccination. Although there have been barrier measures since the start of the pandemic, the morbidity rate is still increasing [6]. It is therefore urgent to find other effective strategies to overcome this disease. According to WHO, the COVID-19 vaccine remains the best weapon to effectively fight the pandemic. As of 23 December 2021, the WHO approved Oxford–AstraZeneca, Johnson and Johnson, Pfizer–BioNTech, Moderna, Sinopharm BIBP, CoronaVac, Janssen, Covaxin, and Novavax vaccines for emergency use worldwide including in low and middle income countries [7,8]. The COVID-19 Vaccines Global Access, abbreviated as COVAX, a worldwide initiative aimed at equitable access to COVID-19 vaccines, tests and therapies directed by the GAVI vaccine alliance, the Coalition for Epidemic Preparedness Innovations (CEPI), and the World Health Organization (WHO), alongside key delivery partner UNICEF, provided vaccines to the developing countries [9,10].

A study on COVID-19 vaccine hesitancy assessment among U.S. medical students conducted on 26 December 2020 was conducted; the majority of participants had positive attitudes towards vaccines and agreed that they would likely be exposed to COVID-19; however, only 53% indicated they would participate in a COVID-19 vaccine trial and 23% were unwilling to take a COVID-19 vaccine immediately [3]. Another study on vaccine reluctance in faculties of health sciences, dentistry and medicine at the University of Malta vis à vis to flu and the new COVID-19 vaccination on 12 November 2020 was made and revealed that the response rate was 23% (*n* = 852) [11]. The percentages of those who received the flu shot last year and will receive it this year have increased at all ages, highest for medicine and universities; for COVID-19 vaccination, little probable, undecided and likely to take were 30.5/25.3/44.02%, respectively. Concerns about the COVID-19 vaccination were related to insufficient knowledge about such a vaccine and potential long-term side effects [11]. Likewise, a study was conducted on the intention of nurses to accept the 2019 coronavirus disease vaccination and change in intention to accept the seasonal flu vaccination during the COVID-19 pandemic in China on the 21 October 2020. It revealed that most nurses went from refusing vaccination to hesitating or accepting than those who went accepting to hesitating or refusing. A total of 40% of participants intended to accept vaccination against COVID-19; these participants were in the private sector. Reasons for refusing and reluctance to take COVID-19 vaccination included “suspicion about efficacy, efficacy and safety”, “deem it unnecessary” and “no time to take it” [12].

Although Cameroon launched its vaccination efforts against COVID-19 in March 2021 with two type of vaccines (Sinopharm and AstraZeneca), only about 5% of the eligible population was vaccinated as of 18 November 2022, placing the country well behind the global target of achieving 70% vaccination coverage by the end of the year [13]. To close the vaccine equity gaps and achieve a broader population coverage goal by the end of the year, representatives from the WHO, UNICEF, GAVI, USAID and Africa CDC came together to advocate for top government leaders, and religious and civil society organizations on the need to support the Ministry of Public Health of Cameroon in its vaccination program against COVID-19 [13]. The anti COVID-19 vaccine acceptance is not shared by all social strata, negatively impacting its implementation. In Cameroon, the epidemiological data on the comparison between urban and rural areas concerning vaccine acceptance were scarce. This study was conducted prior the wide Cameroon’s national vaccination campaign to assess the new anti COVID-19 vaccine awareness in the urban and rural areas in order to contribute more efficiently for anti COVID-19 vaccine implementation.

## 2. Materials and Methods

### 2.1. Study Design, Location, Population and Period

A cross-sectional, descriptive and analytical study was carried out within a six-month period, from March 2021 to August 2021. Our sample was constituted essentially of ≥21 years old unvaccinated individuals residing in urban and rural areas in Cameroon.

### 2.2. Study Setting

A cluster sampling at many degrees was performed by randomly and successfully selecting two mains regions of Cameroon (Littoral and West regions). For that purpose, one populated city (urban area) and village (rural area) were included per region. In the littoral region, Douala (town) and Loum (village) were randomly selected, whereas it was Dschang (town) and Mbouda (village) for the west region. By definition, the choice of urban and rural areas per region was based on geography, population density, social amenities, infrastructure and education facilities and industrialization. In each town/village, streets were first randomly selected via pin through on the map, households were then randomly selected via systematic random sampling using dice throw for each street (the number revealed by the dice represented the number of households to jump between included ones) and a single adult or ≥18 years old individual was finally selected via simple random sampling in each household. Participants were interviewed on their socio-demographic characteristics, knowledge, social behaviors and thoughts relative to COVID-19 disease and vaccine using a structured, pretested and language-adapted questionnaire. All questionnaires with lack of information were excluded.

### 2.3. Data Management, Ethical Considerations

Data were compiled in an Excel spreadsheet and analyzed using Epi info 7.0 software (Center for Disease Control and Prevention, Atlanta, GA, USA) and considering a 95% confidence level. Figures were transposed into Excel 2016 spreadsheet. Percentages were compared using the chi-square test and logistic regression bivariate analyses were carried out with a considered margin of error of 5%. Thus, significant associations were considered when the *p*-value was less than or equal to 0.05.

An ethical approval from the institutional Ethics Committee of the University of Douala (N° 3070CEI-Udo/05/2022/M) and all subsequent administrative authorizations were obtained notably from the sub-divisional officers and mayors of the concerned towns and villages. Free and informed consent of respondents were obtained before each interview.

## 3. Results

### 3.1. Respondent Identification

Throughout this present study, 1056 individuals were included. However, three of them misfiled the questionnaire and were thus excluded. Among the 1053 respondents, 611 were residing in urban areas (396 in Douala and 208 in Dschang) and 442 in rural areas (215 in Loum and 234 in Mbouda). Table 1 summarises the distribution of respondents depending on their socio-demographic characteristics.

Table 1 shows that respondents of the group age < 30 years were the most represented 69.52% (732/1053), both in the urban and the rural areas. The mean age of the respondents was 29.30 ± 6.47 in urban and 28.95 ± 7.00 in rural. For the marital status, the sex, the profession, the single, the female sex and the profession with the resourceful ones represented the majority both in the rural and urban areas. In the two areas, the primary level and the Christian community were the greatest in both groups.

### 3.2. Knowledge of the Respondents on COVID-19 and Vaccination

Table 2 presents the distribution of respondents according to their knowledge.

Table 2 shows that the majority of respondents in the urban and rural areas knew about the existence of COVID-19 [97.55% (596/611) and 85.07% (376/422), respectively)], the existence of an anti-COVID-19 vaccine [97.22% (594/611) and 82.58% (365/422)], respectively)] and that this vaccine could be effective [68.09% (416/611) and 58.60% (259/442)], respectively)]. Nevertheless, the proportions were significantly higher in urban areas than rural areas. Similarly, the proportion of respondents who knew that we can be infected several times by COVID-19 was significantly greater in urban areas; 82.16% (502/611) than in rural areas; 36.65% (162/442). Table 2 also highlights the fact that fewer respondents in both urban and rural areas knew neither the highlighted modes of transmission of COVID-19 [31.26% (191/611) and 33.94% (15/442), respectively] nor the symptoms of this disease [4.91% (30/611) and 23.98% (106/442), respectively] and the possibility of recovery from COVID-19 without treatment [38.30% (234/611) and 28.05% (124/442), respectively]. It was observed that the percentage of respondents did not know COVID-19 symptoms was significantly lower in urban areas than in rural areas (*p* < 0.0001). Oppositely, the percentage of respondents who knew that there was a possibility of recovery from COVID-19 without treatment was significantly greater in urban areas compared to rural areas.

### 3.3. Acceptance of the COVID-19 Vaccine by the Populations

The distribution of respondents according to their acceptance of the COVID-19 vaccine is summarised in Table 3.

It appears from Table 3 that the majority both in the urban and the rural population were neutral for the reception of the COVID-19 vaccine [44.84% (274/611) and 51.58% (228/442), respectively]. Despite the relatively smaller percentage of respondents who agreed to accept anti-COVID-19 vaccine in both areas, it was significantly greater in the urban areas than in the rural areas (*p* = 0.0047). The most frequent reason for reluctance to anti COVID-19 vaccination was the fact that the vaccine may be not safe due to the speed of synthesis in both urban and rural areas (52.86% and 35.75%, respectively). Moreover, the reason based on the fact that the vaccine was to give a disease was significantly more frequent in rural areas (35.07%) than in urban areas (8.84%).

### 3.4. Determinants of Acceptance of Vaccination against COVID-19

Table 4 and Table 5 show the analysis of the probable associated factors to the acceptance of the anti-COVID-19 vaccine in univariate and multivariate logistic regression, respectively.

Overall, it appears from Table 4 that only the profession (trader) was associated with the acceptance of anti-COVID-19 vaccine acceptance in urban areas (*p* = 0.0046). Meanwhile, both the profession (trader) and the study level (Superior) were statistically associated with vaccine acceptance in rural areas [*p* < 0.0001 and *p* = 0.0001, respectively).

At this purpose, it was observed that the odds of being traders among respondents intending to accept anti-COVID-19 vaccine was, respectively, 0.54 folds and 0.21 folds the odds of being resourceful in urban and rural areas. Moreover, the odds of being at the superior educational level among respondents ready to accept anti-COVID-19 vaccine was 2.57 folds higher as compare to those with a null educational level.

## 4. Discussion

In order to contribute in the prevention and the control of the current COVID-19 pandemic and increase the adherence of the population to the vaccination, the present study was conducted in order to determine the epidemiology of the acceptance of vaccine against COVID-19 in Cameroon urban and rural areas. To carry out this study, we collected data with a structured, pretested and language adapted questionnaire.

Our sample consisted of 1053 individuals. From the applied methodology, it resulted that out of 1053 participants, a preponderance was noted toward females (84.81%), single respondents (61.73%), resourceful respondents (56.60%), primary level respondents (41.41%), and Christians (93.73%). This globally describes the most frequent socio-demographics in Cameroon. Moreover, we noticed more participants in urban than in rural area (611 vs. 442). A similar result was observed in a study conducted nationwide in Cameroon from 2020 to 2022 on factors driving COVID-19 vaccine hesitancy in Cameroon and their 9mplications for Africa, with 56.3% (3787/6732 from urban 43.7% vs. (2945/6732) from rural [14]. In contrast to urban settings, many people in rural areas were not really interested in such studies because of their misbelief, past history and lack of education [14]. Furthermore, the mean age of the respondents in this study was quite similar in both settings (29.30 ± 6.47 in urban and 28.95 ± 7.00 in rural) with subjects aged <30 years old (69.52%) most represented. A study conducted on gender and COVID-19 vaccine disparities in Cameroon reported quite a similar mean age of study participants, with participants aged <34 years most represented at 62.6% (153/249) [15]. In fact, as all African countries, Cameroon is made up of a young working population [16]. Additionally, as compared to young population, a higher reluctance to participate to the COVID-19 related activities was globally observed in older adults [16,17].

Despite the fact that the percentage of respondents who did not know COVID-19 symptoms were significantly lower in urban than in rural areas, the proportion of respondents aware of the existence of COVID-19 pandemic in contrast was globally high in both areas with slightly greater proportion in the urban area [97.55% (596/611) urban and 85.07% (376/422) rural]. This result was close to the one reported in the towns of Douala and Bangangté in 2020, where 71.6% declared that COVID-19 existed [18]. In fact, this was the result of cumulative efforts between WHO, non-government organisations, and governments including wide sensitization and financial support in response against the pandemic spread worldwide [19]. In addition, as COVID-19 evolved at the time of new technologies, the information about the deadly epidemic rapidly and widely spread through mass media, internet, from school or conferences and person to person.

In this study, 38.65% (407/1053) intended to accept anti-COVID-19 vaccine. This vaccine acceptance rate was relatively lower than that reported by the WHO (72.4%) [4], and Roman et al. (44.2%) [20], Lazarus et al. 79.1% [21]. Several factors could explain that relative low acceptance rate in our study such as mistrust of government and health authorities, concerns about vaccine safety and efficacy, some obstacles to effective vaccine science communication for lay audiences, public perceptions about that new COVID-19 variants can be possibly less severe, internet driven misinformation and fake news [22]. Of note, there were still a relative high proportions of the population who did not even believe neither in the existence of COVID-19 7.63% (81/1053) and anti-COVID vaccination 8.9%(94/1053) nor in the effectiveness of the vaccine 35.9%(378/1053). Taken together, all these data strengthened the need of a continuous education and sensitization of the population about the importance and safety of anti-COVID-19 vaccine nationwide.

Nevertheless, the anti-COVID vaccine acceptance rate was greater in urban than in rural areas (42.55% vs. 33.26%; *p* = 0.0047). These results were similar to those found in 2022 by Verena Barbieri et al. in Italy [23]. This similitude can be explained by the fact that reservations regarding COVID-19 vaccination strongly depend on the sociocultural characteristics of the study population. In addition to knowledge about the safety and efficacy of vaccines, fear of side effects, and level of information about the disease and vaccination, religious attitudes may play an important role in rural areas [24]. Reduced interest in information and a greater lack of trust in institutions and information sources among rural residents were also confirmed.

Upon the answers concerning the reason why they were not accepting the COVID-19 vaccine, the most represented answer was “the vaccine may not be safe/efficient enough regarding the rapid speed of synthesis and approval for emergence use by WHO”. Many studies, in line with our inquiry, already incriminated concerns about vaccine safety and efficacy as one of the major reason of vaccine hesitancy [21,22,25]. This can be explained by the fact that, intentional or not, misrepresentation and misinformation can derail progress in COVID-19 vaccination coverage, particularly if audiences choose not to seek COVID-19 information from official sources, such as WHO, the US Centers for Disease Control and Prevention or medical professional associations. These high-credibility sources of information face the additional challenges of pandemic fatigue or distress that may demotivate one to follow recommended protective behaviors and, among some communities, the challenge of low trust toward such institutions [26]. In fact, failure to convey clearly and consistently information to lay audiences during the current pandemic may have confused audiences, eroded confidence in the science and reduced vaccine acceptance [26]. With new technologies, self-digital health literacy through social media could have exacerbated this phenomenon [22].

Upon the completion of our study, the acceptance of anti-COVID-19 vaccine was significantly associated with the university level of education (*p* = 0.0001) in the rural area. This was obviously due to the easier aptitude to access good quality information, as a result of which the respondents were hence most aware on correct anti-COVID-19 vaccination issues [18]. Moreover, trader was the profession with the lowest rate of anti-COVID-19 vaccine acceptance in both urban and rural areas (*p* = 0.0046 and *p* < 0.0001, respectively). This result indicates the most reluctant socio-professional category to anti-COVID-19 vaccination campaigns. Nevertheless, basic reasons for this reluctance is still unknown, hence suggesting the initiation of researches targeting traders in urban and rural areas in order to better investigate this phenomenon. This can represent an important baseline to mitigate reluctance to anti-COVID-19 vaccination in a general issue.

## 5. Limitations

During this study, eligible participants in some household were skeptical and reluctant to participate due to misrepresentation and misinformation about COVID-19 and anti-COVID 19 vaccines. Moreover, since it was a face-to-face interview, eligible people absent from their household or who travelled abroad could not be reached during the survey period. Lastly, psychological patterns linked to vaccine hesitancy were general and non-specific, and thus, there is a need for more specific and further study on the subject.

## 6. Conclusions

This research showed that knowledge related to COVID-19 and its vaccine was better in urban areas compared to rural areas. Additionally, the acceptance intention was significantly more frequent in urban areas than in rural areas. Moreover, the significant determinants of anti-COVID-19 acceptance were the level of education and profession in the rural areas, and only the profession in the urban areas.

## Figures and Tables

**Table 1 vaccines-11-00625-t001:** Distribution of respondents according to their socio-demographic characteristics.

Variables	Urban (N = 611)	Rural Area (N = 442)	*p*-Value
	*n* (%)	*n* (%)	
Age			
<30	420 (68.74)	312 (70.59)	
[30.0–40.0]	119 (19.80)	85 (20.14)	0.70
>40	72 (11.46)	45 (9.28)	
Marital status			
Single	373 (61.05)	277(62.67)	
Divorced/Widow	12 (1.96)	3 (0.68)	0.98
Married	226 (36.99)	162 (36.65)	
Sex			
Female	519 (84.94)	374 (84.62)	0.97
Male	92 (15.06)	68 (15.38)	
Profession			
Trader	119 (19.48)	87 (19.68)	
Student	24 (3.93)	64 (14.48)	0.99
Resourceful *	384 (62.85)	212 (47.96)	
Others *	841 (3.75)	79 (17.87)	
Level of education			
None	64 (10.47)	157 (35.52)	
Primary	271 (44.35)	165 (37.33)	0.87
Secondary	99 (16.20)	36 (8.14)	
Superior	177 (28.97)	84 (19.00)	
Religion			
Animist	2 (0.33)	2 (0.45)	
Christian	574 (93.94)	413 (93.44)	0.93
Muslim	35 (5.73)	27 (6.11)	

* Resourceful: Hustler, Person different from student or trader, without a full-time job and low income/month; others: teacher, taximen, housewife, driver, carpenter, etc.

**Table 2 vaccines-11-00625-t002:** Distribution of respondents according to their knowledge on the existence and transmission routes of COVID-19.

Variables	Urban (N = 611)	Rural (N = 442)	p-Value
	n (%)	n (%)	
COVID-19 exists			
No	15 (2.45)	66 (14.93)	
Yes	596 (97.55)	376 (85.07)	<0.0001
Modes of transmission			
No	420 (68.74)	292 (66.06)	
Yes	191 (31.26)	150 (33.94)	0.3602
Awareness of COVID-19 symptoms:			
No	581 (95.09)	336 (76.02)	
Yes	30 (4.91)	106 (23.98)	<0.0001
We can recover from COVID-19 without treatment			
No	377 (61.70)	318 (71.95)	
Yes	234 (38.30)	124 (28.05)	0.0005
We can be infected several times by COVID19			
No	109 (17.84)	280 (63.35)	
Yes	502 (82.16)	162 (36.65)	<0.0001
There is a vaccine against COVID-19			
No	17 (2.78)	77 (17.42)	
Yes	594 (97.22)	365 (82.58)	<0.0001
Anti COVID-19 Vaccination can be effective			
No	195 (31.91)	183 (41.40)	
Yes	416 (68.09)	259 (58.60)	0.0016

**Table 3 vaccines-11-00625-t003:** Distribution of the respondents according of anti-COVID-19 vaccine.

Variables	Urban (N = 611)	Rural (N = 442)	*p*-Value
	*n* (%)	*n* (%)	
Would you agree to receive COVID-19 vaccine?			
Neutral	274 (44.84)	228 (51.58)	Ref
Agree	260 (42.55)	147 (33.26)	0.0047
Disagree	77 (12.60)	67 (15.16)	0.8137
Reasons for not accepting the COVID-19 vaccine			
It is a negligible threat disease and can be cured	48 (7.86)	27 (6.11)	Ref
I do not have enough information about the vaccine	186 (30.44)	102 (23.08)	0.9251
It is to give us a disease	54 (8.84)	155 (35.07)	<0.0001
The vaccine may be not safe due to the speed of synthesis	323 (52.86)	158 (35.75)	0.5902

**Table 4 vaccines-11-00625-t004:** Factors to acceptance of the vaccine against COVID-19 in a univariate logistic regression.

Variables	Vaccine Acceptance (Urban Area)		Vaccine Acceptance (Rural Area)	
	%(*n*/N)	*p*-Value, OR (95% CI)	%(*n*/N)	*p*-Value, OR (95% CI)
Age				
<30	40.00(168/420)	Ref	34.94(109/312)	Ref
[30.0–40.0]	47.11(57/121)	0.16, 0.34 (0.89–2.01)	31.46(28/89)	0.54, 0.85 (0.52–1.42)
>40	50.00(35/70)	0.11, 0.50 (0.90–2.49)	24.39(10/41)	0.18, 0.60 (0.28–1.27)
Marital status				
Single	39.41(147/373)	Ref	31.77(88/277)	Ref
Married	44.69(101/226)	0.20, 1.24 (0.89–1.74)	36.42(59/162)	0.32, 1.23 (0.82–1.85)
Sex				
Féminin	41.81(217/519)	Ref	33.69(126/374)	Ref
Masculin	46.74(43/92)	0.37, 1.22 (0.78–1.91)	30.88(21/68)	0.65, 0.88 (0.50–1.54)
Profession				
Resourceful	47.66(183/384)	Ref	38.68(82/212)	Ref
Trader	32.77(39/119)	0.004, 0.54 (0.35–0.82)	11.49(10/87)	<0.000, 0.21 (0.10–0.42)
Student	41.67(10/24)	0.57, 0.78 (0.34–1.81)	45.31(29/64)	0.34, 0.61 (0.31–0.752)
Others	33.33(28/84)	0.058, 0.55 (0.33–0.90)	32.91(26/79)	0.3657, 0.78 (0.45–1.34)
Level of education				
None	43.75(28/64)	Ref	23.57(37/157)	Ref
Primary	35.06(95/271)	0.19, 0.69 (0.40–1.21)	42.24(73/165)	0.1522, 1.54 (0.85–2.77)
Secondary	46.46(46/99)	0.73, 0.12 (0.59–2.10)	27.78(10/36)	0.5960, 1.25 (0.55–2.82)
Superior	51.41(91/177)	0.29, 1.36 (0.77–2.42)	32.14(27/84)	0.0001, 2.57 (1.59–4.16)

**Table 5 vaccines-11-00625-t005:** Factors to acceptance of the vaccine against COVID-19 in a multivariate regression analysis.

Variables	Vaccine Acceptance (Rural Area)		
	%(*n*/N)	*p*-Value	Adjusted OR (95% CI)
Profession			
Resourceful	38.68(82/212)	Ref	Ref
Trader	11.49(10/87)	0.0002	0.23(0.11–0.50)
Student	45.31(29/64)	0.57	1.18(0.66–2.10)
Others	32.91(26/79)	0.23	0.71(0.40–1.24)
Level of education			
None	23.57(37/157)	Ref	Ref
Primary	42.24(73/165)	0.41	1.30(0.70–2.42)
Secondary	27.78(10/36)	0.22	1.72(0.72–4.12)
Superior	32.14(27/84)	0.0167	1.87(1.12–3.13)

## Data Availability

All data are available upon request.

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
