# Peer review of "Epidemiology of the Acceptance of Anti COVID-19 Vaccine in Urban and Rural Settings in Cameroon"

_vaccines, 2023, doi:10.3390/vaccines11030625_

Round 1

Reviewer 1 Report

The authors have made an interesting attempt on “Epidemiology of the acceptance of anti covid-19 vaccine in urban and rural settings in Cameroon.” The manuscript is interesting; however, the authors need to justify the scientific writing manuscript. Some of the general comments are provided below:

1.     Did authors conducted a pilot study to validate the clarity and comprehensibility of survey questions, if yes, what is the test score of Cronbach's alpha.

2.     What is the representative minimum sample size for Cameroon population required for statistical analysis?

3.     The authors should mention the types of vaccines available in Cameroon.

4.     It would be interesting to know which type of vaccine is more acceptable among rural and urban people.

5.     The authors include the limitations of the study.

6.     The authors should discuss the reason that why the percentage of respondents who did not know COVID-19 symptoms was significantly lower in urban areas than in rural areas.

7.     The authors should explain the term “resourceful” used in the professions.

8.     Why there is huge gap between male and female participants (15% to 85%)?

9.     There are some typo errors, for example line 101, change “news” to “new”.

Author Response

The authors have made an interesting attempt on “Epidemiology of the acceptance of anti covid-19 vaccine in urban and rural settings in Cameroon.” The manuscript is interesting; however, the authors need to justify the scientific writing manuscript. Some of the general comments are provided below:

  1. Did authors conducted a pilot study to validate the clarity and comprehensibility of survey questions, if yes, what is the test score of Cronbach's alpha.

Answer:  Yes dear reviewer, as mentioned in the method, the questionnaire was pretested and language adapted. Moreover, it was always filled in presence of a research team member.

  1. What is the representative minimum sample size for Cameroon population required for statistical analysis?

Answer: Thanks for that question dear reviewer. In fact, the minimum sample size for statistical analysis is 30, but it is always recommended to work on the largest sample size possible as it should be more representative of the study population and also increase the power of the study.

  1. The authors should mention the types of vaccines available in Cameroon.

Answer: Thanks dear reviewer. We added it in the introduction.

  1. It would be interesting to know which type of vaccine is more acceptable among rural and urban people.

Answer: Thanks dear reviewer. Cameroon’s national vaccination campaign was launched on April 2021 with only two types of vaccines: Sinopharm and AstraZeneca and targeting a really restricted eligible population (eventually Health personnel and subject aged >50years old). Giving the high reluctance to vaccine observed at the beginning (only about 5% of target population successfully vaccinated after 5months) we found interesting to assess knowledge of population far before vaccination, independently of the type of the vaccine available.

  1. The authors include the limitations of the study.

Answer: Thanks dear reviewer. We added it

  1. The authors should discuss the reason that why the percentage of respondents who did not know COVID-19 symptoms was significantly lower in urban areas than in rural areas.

Answer: Thanks dear reviewer for that point. We added it in the discussion section

  1. The authors should explain the term “resourceful” used in the professions.

Answer: done dear reviewer, Thanks.

  1. Why there is huge gap between male and female participants (15% to 85%)?

Answer:  Apart of the Cameroon demographic report where women are more represented, It could also be explained by the fact that women adhere more easily on health related matters than men.

  1. There are some typo errors, for example line 101, change “news” to “new”.

Answer: Corrected. Thanks reviewer,

Submission Date

03 February 2023

Date of this review

09 Feb 2023 08:50:57

Reviewer 2 Report

Estimated Authors,

I've read with great interest the present paper from Djuikoue et al on the acceptance of SARS-CoV-2 vaccine in rural areas of Camerun. The paper is not quite innovative in its results, but its significance is substantial, as it shares a piece of information about a geographical area that has been scarcely inquired by similar studies, providing information about the factors that either promote or impair vaccine acceptance.

Still, I've a couple of suggestions, most of them quite formal ones:

- an univariate comparison (Table I) between Urban and Rural Area would provide further interesting information that could contribute to the discussion section. For example, it is quite consistent with previous studies that the education level in rural areas is lower than that from urban ones. 

- Why did you chose following age groups: 18 - 30 (quite conventional, ok), 30 - 42 and > 42? a cut off at 42 years is quite unusual; please explain

- across the tables there are a series of French typos: for example, Féminin for females; please double check

- Table II : "the symptoms of COVID-19 are ..." but the sentence is incomplete, as the reported answer is only yes vs. no, while the question is clearly designed through the description of symptoms. Please rework.

Author Response

Estimated Authors,

I've read with great interest the present paper from Djuikoue et al on the acceptance of SARS-CoV-2 vaccine in rural areas of Camerun. The paper is not quite innovative in its results, but its significance is substantial, as it shares a piece of information about a geographical area that has been scarcely inquired by similar studies, providing information about the factors that either promote or impair vaccine acceptance.

Still, I've a couple of suggestions, most of them quite formal ones:

- an univariate comparison (Table I) between Urban and Rural Area would provide further interesting information that could contribute to the discussion section. For example, it is quite consistent with previous studies that the education level in rural areas is lower than that from urban ones.

Answer: Thanks reviewer, we did it.

 - Why did you chose following age groups: 18 - 30 (quite conventional, ok), 30 - 42 and > 42? a cut off at 42 years is quite unusual; please explain

Answer: Dear reviewer, we revised the age range as follow: <30, 3[0-40], >40

- across the tables there are a series of French typos: for example, Féminin for females; please double check

Answer: Thanks reviewer, we did it.

- Table II : "the symptoms of COVID-19 are ..." but the sentence is incomplete, as the reported answer is only yes vs. no, while the question is clearly designed through the description of symptoms. Please rework.

Answer: Thanks reviewer, we did it.

Submission Date

03 February 2023

Date of this review

20 Feb 2023 22:06:31

Reviewer 3 Report

The authors presented the results of a questionnaire survey asking about acceptance and attitude toward SARS-CoV-2 vaccination in Cameroon. The results are of relevance to developing public measures to improve vaccine coverage. In particular, the comparison between urban and rural areas is helpful in modifying public measures.

1) In the introduction section, would you show the COVID-19 vaccine coverage in Cameroon at the beginning of the questionnaire survey, that is, in March 2021? The readers would like to know the social background in Cameroon -i.e, when the vaccination campaign and mass vaccination were started in Cameroon, and how the vaccination coverage was elevated in Cameroon.

2) The analyses were done with the classification of urban or rural areas in this study. In the introduction section, please briefly mention previous findings or authors’ hypotheses for classifying urban and rural areas.

3) If available, please add the estimated population density of Douala, Loum, Dschang, and Mbouda. The definition of urban and rural areas in this study is very important.

4) Despite the efforts of random sampling, the differences in profession and level of education between urban and rural areas might contribute to the results shown in Tables 2 and 3. The authors should mention the limitations of this study in the discussion section.

5) Multivariate regression analysis shown as an adjusted odds ratio would be appreciated to demonstrate in Table 4, in addition to the univariate analysis, to understand the overall effects of individual variables.

Author Response

The authors presented the results of a questionnaire survey asking about acceptance and attitude toward SARS-CoV-2 vaccination in Cameroon. The results are of relevance to developing public measures to improve vaccine coverage. In particular, the comparison between urban and rural areas is helpful in modifying public measures.

  • In the introduction section, would you show the COVID-19 vaccine coverage in Cameroon at the beginning of the questionnaire survey, that is, in March 2021? The readers would like to know the social background in Cameroon -i.e, when the vaccination campaign and mass vaccination were started in Cameroon, and how the vaccination coverage was elevated in Cameroon.

Answer: dear reviewer we added that really interesting point.

  • The analyses were done with the classification of urban or rural areas in this study. In the introduction section, please briefly mention previous findings or authors’ hypotheses for classifying urban and rural areas.

Answer: Okay dear reviewer we added it

  • If available, please add the estimated population density of Douala, Loum, Dschang, and Mbouda. The definition of urban and rural areas in this study is very important.

Answer: Thanks dear reviewer. In fact, by definition, the choice of urban and rural areas per region was based on criteria like geography, population density, social amenities, infrastructure and education facilities, industrialization.

  • Despite the efforts of random sampling, the differences in profession and level of education between urban and rural areas might contribute to the results shown in Tables 2 and 3. The authors should mention the limitations of this study in the discussion section.

Answer: Thanks dear reviewer, we added the limitations of the current study as requested. By the way, concerning profession and level of education between urban and rural areas the univariate analysis of table I didn’t really reveal any statistically significant difference.

  • Multivariate regression analysis shown as an adjusted odds ratio would be appreciated to demonstrate in Table 4, in addition to the univariate analysis, to understand the overall effects of individual variables.

Answer: Thanks dear reviewer, we added it

Submission Date

03 February 2023

Date of this review

20 Feb 2023 09:14:20

Round 2

Reviewer 1 Report

The authors have addressed my queries,  now the manuscript is acceptable for publication.